# Synergistic effects of anti-PDL-1 with ablative radiation comparing to other regimens with same biological effect dose based on different immunogenic response

**Maedeh Alinezhad[1], Mohsen Bakhshandeh[2], Elham Rostami[3], Reza Alimohamadi[1], Nariman Mosaffa[1], Seyed Amir Jalali**[1]*

**1** Department of Immunology, Medical School, Shahid Beheshti University of Medical Sciences, Tehran, Iran, **2** Department of Radiology Technology, Allied Medical Faculty, Shahid Beheshti University of Medical Sciences, Tehran, Iran, **3** Department of Immunology, School of Medicine, Mashhad University of Medical Sciences, Mashhad, Iran

* jalalia@sbmu.ac.ir, jalali5139@yahoo.com

**Data Availability Statement:** Data are available at https://doi.org/10.7910/DVN/NXBKZG.

## Abstract

### Introduction

Irradiation can induce multiple inhibitory and stimulatory effects on the immune system. In recent studies, it has been noted that administration of radiation with various doses and fractionation plans may influence on immune responses in microenvironment of tumor. But in radiobiology, the Biologically Effective Dose (BED) formula has been designed for calculating isoeffect doses in different regimens of daily clinical practice. In other words, BED has also been used to predict the effects of fractionation schedules on tumor cells.

### Methods

In our study, three different regimens with BEDs of 40 gray (Gy) were analyzed in BALB/c mice. These included conventional fractionated radiotherapy (RT) (3Gyx10), high-dose hypofractionated RT (10Gyx2), and single ablative high-dose RT (15Gyx1).

### Results

As BED predicts, all three similarly decreased tumor volumes and increased survival times relative to controls, but after high dose exposure in ablative group, the expression of IFNγ was increased following high infiltration of CD8 cells into the tumor microenvironment. When anti-PDL-1 was combined with RT, single ablative high-dose radiation enhanced antitumor activity by increasing IFNγ in tumors and CD8$^+$ tumor-infiltrating lymphocytes; as a result, this combining therapy had enhanced antitumor activity and lead to control tumor volume effectively and improve significantly survival rate and finally the recurrence of tumor was not observed.

**Funding:** The present article was financially supported by a grant from the research council of Shahid Beheshti University of Medical Science in Iran (Grant No. 10588). The funders had no role in study design, data collection and analysis, decision to publish, or preparation of the manuscript.

**Competing interests:** The authors have declared that no competing interests exist.

**Abbreviations:** BED, biological effect dose; Gy, gray; RT, radiotherapy; ICB, immune checkpoint blockers; ICD, immunogenic cell death; DAMPS, Damage-associated molecular patterns; HMGB1, high mobility group box 1 protein; TME, tumor microenvironment; PDL1, programmed death-ligand 1; LQ model, linear quadratic model; CT26, colon carcinoma cells; FBS, fetal bovine serum; RPMI media, Roswell Park Memorial Institute media; TTE, time to endpoint; TGD, tumor growth delay; TILs, tumor infiltrating lymphocytes; IFNγ, Interferon gamma; FOXP3, forkhead box P3; LTS, long term surviving.

## Conclusion

Results show distinct radiation doses and fractionation schemes with same BED have different immunogenic response and these findings can provide data helping to design regimens of radiation combined with immune checkpoint blockers (ICBs).

## Introduction

The main purpose of irradiation is to eliminate tumor cells through DNA damage in cancer therapy, resulting in apoptosis, necrosis, and autophagy, to ultimately prevent tumor growth [1]. Studies indicate radiotherapy (RT) can stimulate the immune system by inducing immunogenic cell deaths (ICDs) that characterized by the release of Damage-associated molecular patterns (DAMPS), high mobility group box 1 protein (HMGB1), and ATP, and by modifying the tumor microenvironment (TME) [2]. Irradiation has multiple inhibitory and stimulatory effects on the immune system such as increased CD4$^+$, CD8$^+$ and Treg Cells infiltration into TME [3, 4] Some regimens of radiation like ablative RT was associated with increased infiltration of exhausted CD8+ T cells into the tumor, which induced radiation resistance. [5, 6]. The combination of radiotherapy and Immune checkpoint blockers therapy can reduce these inhibitory effects by stimulating immune cells and enhancing the response to radiation. Preclinical studies showed that RT increased PDL-1 expression on tumor cells [7, 8], and anti-PDL1 (αPDL-1) mAb combined with radiation had a synergistic effect on immune response induction dependent IFN-γ producing CD8 T cells activations. [9, 10].

It has also been noted that radiation doses and fractionation schemes distinctly impact both the host immune system and the tumor cell's immunogenicity [11, 12]. Many studies are ongoing to characterize their inhibitory and stimulatory effects [13], and to improve their effects by combinations of RT and immunotherapy [14, 15].

In radiobiology, the Biologically Effective Dose (BED) formula is used to calculate isoeffect doses in various regimens of daily clinical practice. BED has also been used to predict the effects of fractionation changes on tumors according to the linear quadratic (LQ) model that describes cell response to irradiation [16].

A recent meta-analysis study shows when the BED increases, the occurrence rate of abscopal effects also increases due to immunological responses in preclinical models [17]. However, studies have not yet determined whether different regimens with the same BED (when the biological effects of the regimens are equal) have different immunogenic effects, whether they affect tumor recurrence associated with immunological responses, In addition, can these regimens respond differently to combination therapy? Because it can improve the quality of immune responses into TME by use of modification in the dose and number of fractions radiation to the extent that the radiation clinic limitations and technology of the radiation devices allow us. Also it can lead to memory responses, which prevents tumor recurrence that is a common problem after treatment.

To address these issues, we aimed to evaluate the synergistic effect of RT combined with immunotherapy using αPDL-1 in three different RT regimens with the help of modeled tumor BALB/c mice.

## Materials and methods

### Mice and cell lines

CT26 murine colon carcinoma cells (Pasteur Institute of Iran, Tehran, Iran) were cultured in RPMI-1640 media (Cassion, US) enriched with 10% fetal bovine serum (FBS) (Gibco, US),

10,000 IU Penicillin, 10,000 mg/mL Streptomycin, and 1% L-glutamine (Cassion, US) and used at limited passage number. Female BALB/c mice (4 to 6 week old) were obtained from Razi Institute of Iran. The mice, eight weeks old at start time, were randomly selected in 8 groups with at least ten mice in each groups [10]. Mice were purchased with a health report according to Pasteur Institute routine health monitoring program and kept in the animal house under standard controlled conditions. All experiments were approved by the Institutional Ethical Committee and Research Advisory Committee of Shahid Beheshti University of Medical Sciences with code ethic number: IR. SBMU. MSP.REC.1395.457.

## Tumor challenge

For tumor induction, $1 \times 10^6$ CT26 tumor cells were inoculated subcutaneously into the right flank of anesthetized mice as described [18]. Tumor volumes ($mm^3$) were calculated with formula length $\times$ width $\times$ height $\times$ 0.52 that measured by a digital caliper. For tumor re-challenge experiments, long-term surviving (LTS) mice were inoculated with $1 \times 10^6$ CT26 tumor cells in the left flank 90 and 150 days after the initial tumor implantation.

## Calculate BED

To deliver a 40 Gy equivalent total BED in a single dose as ablative radiation, two fractions as hypofraction radiation, or ten fractions as conventional radiation, the dose per fraction was calculated with LQ models. The biological effect dose of 3 regimens radiation including single dose received 15 Gy (15 Gy $\times$ 1), two fractions received 10 Gy per fraction (10 Gy $\times$ 2), and 10 fractions received 3 Gy per fraction (3 Gy $\times$ 10, is approximately equal 40 Gy (The a/b ratio was considerate 10 Gy for soft tissue tumor like CT26 Tumor model) [19].

## Treatment

All mice were irradiated 18 days' post-initial inoculation. At that time the tumors were at least 300–400 $mm^3$. The experimental schedule of ablative radiation therapy was only one fraction of 15 Gy in 18 days' post-initial inoculation. In Hypofraction RT regimens, two fractions of 10 Gy were performed on days 18 and 28 after inoculation. In Conventional RT, ten fractions of 3 Gy were given from day 18 to day 31 for 14 days at a time interval of one day or at most two days. (Fig 1A). Before irradiation the mice were anesthetized by intraperitoneal injection with 100 mg/kg ketamine 10% and 12.5 mg/kg Xylazine 2% (Alfasan, Sofia, Bulgaria). Mice were irradiated using a clinical linear accelerator (6 MV photons, Elekta synergy linear accelerator, Stockholm, SE). Welfare considerations were taken to help mice efforts from minimize suffering and distress. The mice were placed in a modified 50 ml plastic tube, which helped the area of the tumor to be irradiated while keeping the rest of the body outside the RT field. All parts of the body except the irradiation field were protected by a 9-cm-thick lead plate. Radiotherapy was delivered to a 3×3 $cm^2$ field with 5-mm margins at 350 Gy/min with 6 MV X-ray using tangential beam delivery. Super flab Bolus Material of 1.5 cm was placed over the tumor, and the source-to-skin distance was 100 cm. Mice received intraperitoneal 200 µg/ml of PDL1-blocking antibody 10F.9G2 clone (Bio X Cell, NH, USA) on day 18 (the starting radiation day), and on days 21 and 24 in the combined therapy groups.(Fig 2A)

## Measurement of survival factors: *In vivo* study

The time required for a tumor to reach a final volume of 1500 $mm^3$ is called Time to Endpoint (TTE), which is calculated by the formula TTE = [log (end point)-b]/m. A TTE diagram is derived from the linear regression of the log of tumor growth at times that "b" and "m" are the

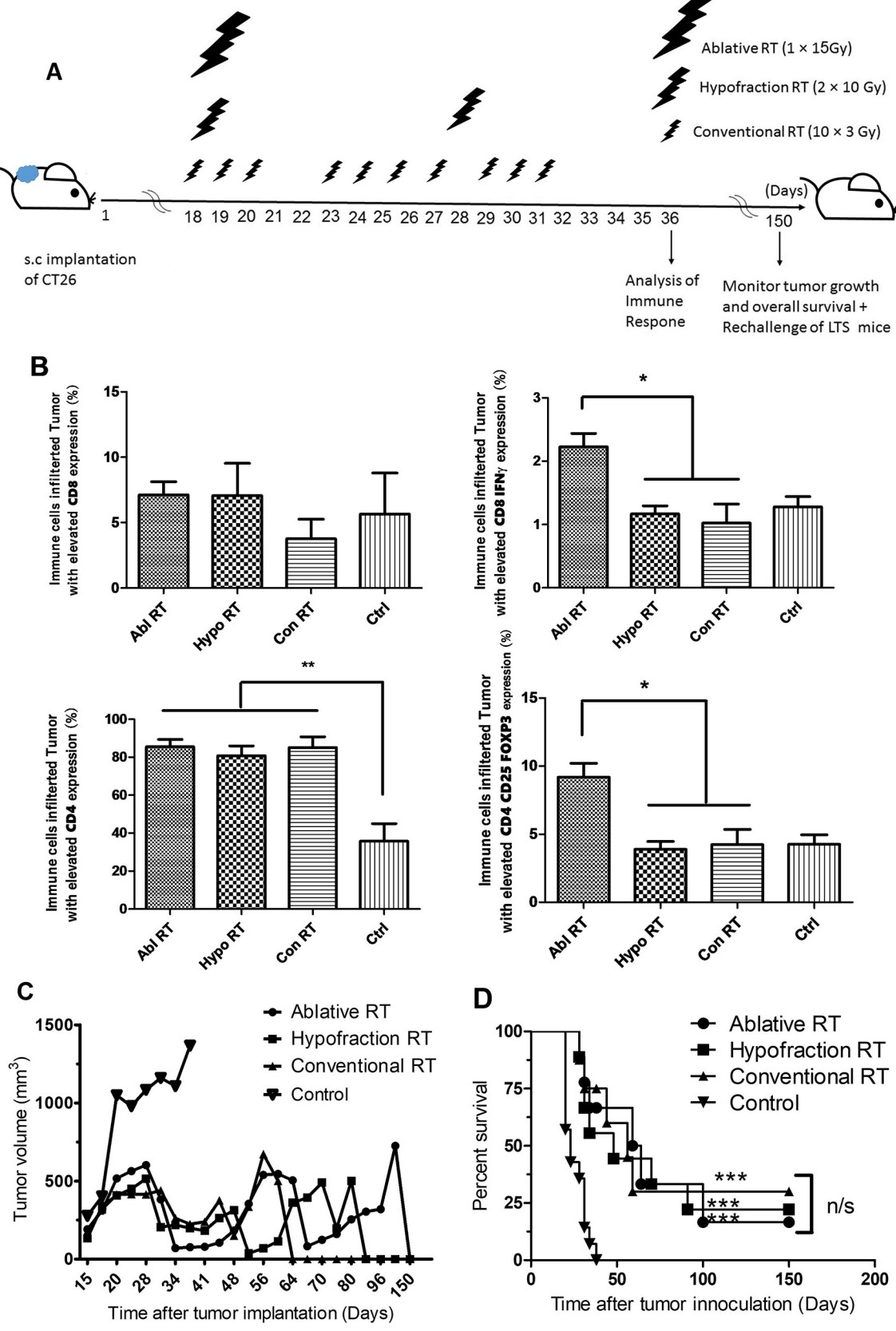

**Fig 1.** Ablative RT comparing to other regimens with same BED increase numbers and ratios of immune cell in the TME but no differences in survival rates or tumor volumes The experimental schedule of radiation therapy in ablative (n = 9), hypofraction (n = 9), and conventional (n = 9) radiation regimens are shown (A). $1 \times 10^6$ CT26 cells were inoculated subcutaneously into the right flanks of

mice on starting day and irradiation of each group began 18 days after initial tumor implantation. In day 36, 3 mice of each group were analyzed for the percentage of immune cells (CD8$^+$, CD8$^+$ IFN$\gamma^+$, CD4$^+$, CD4$^+$ CD25$^+$ FOXP$_3^+$) that infiltrated into the tumor are shown in (B). Data are presented as means ± SDs and analyzed by Tukey's Multiple Comparison Test; (*: P <0.05, **: P < 0.01). Tumor volumes and survival rates are shown respectly in (C) and (D). There is significant different (***: P < 0.001) between RT therapy and Control group (n = 14) and no significant differences in survival rates were seen between mice treated with the different RT regimens. Results were analyzed by the log-rank (Mantel-Cox) Test; n/s: **not** significant.

y-intercept and slope, respectively. The end point criteria were 1) tumor volume became greater than 1500 mm$^3$, 2) body weight decreased 15 percent or more of the initial weight, and 3) health decline or dead. Percent of tumor growth delay (%TGD) was calculated from the ratio of TTE in experimental groups to the control group, or each group as described [20].

### Measurement of infiltrating immune cells in tumor, lymph nodes, and spleen by flow cytometry

After the treatment protocol, 36 days' post-tumor challenge, some mice of groups were first anesthetized with isoflurane and then sacrificed by cervical dislocation. The tumors were minced into small pieces then were incubated with Type I collagenase in RPMI medium 1640 (1:1 ratio) at 37˚C for two hours. Lymph nodes near the tumor and the spleen were also cut into small pieces, minced, pelleted, and washed two times for 5 min with RBC lysis buffer. The cells were filtered through a 70 μm cell strainer (BD Falcon, USA) and then centrifuged at 300g for 10 min. So, the pellets of cells were suspended in flow cytometry staining buffer (phosphate-buffered saline containing 5% FBS) and analyzed by flow cytometry using fluorochrome anti-bodies against CD4 (clone GK1.5), CD8 (clone 53–6.7), CD25 (clone 3C7), Foxp3 (clone 150 D), and IgG1 isotype control (clone MOPC-21) (Biolegend, San Diego, California) [21].

### Measurement of IFN$\gamma$ production in tumor, lymph node, and spleen by flow cytometry

The tumor-infiltrating lymphocytes (TILs) and lymph node and spleen cells were cultured with cell activation cocktail (PMA/Ionomycin with Brefeldin A, Biolegend, San Diego, Californian) for 4 hours, centrifuged at 300g for 10 min, and suspended in flow cytometry staining buffer. Cells were analyzed by flow cytometry for the expression of IFN$\gamma$ (clone XMG1.2), IgG1 isotype control (clone RTK2071), CD8, and CD4. Flow cytometry was performed with a BD FACS Cali-bur flow cytometer (Becton Dickinson, USA) and analysis with FlowJo 7.6.1 software.

### Statistical analysis

The results are presented as means ± standard deviations (SDs) of the means. Statistics were analyzed using the independent t-test, and the post hoc test for one-way ANOVA by GraphPad Prism version 5 (GraphPad Software, San Diego, CA, USA). Survival was analyzed with the log-rank Mantel–Cox test. P values < 0.05 were considered significant.

## Results

### Ablative RT comparing to other regimens with same BED increased numbers and ratios of immune cell in the TME but no differences in survival rates or tumor volumes

The radiation therapy schedules are shown in Fig 1A. The percentage of immune cells that infiltrated to the tumor are shown in Fig 1B. Ablative radiation significantly increased

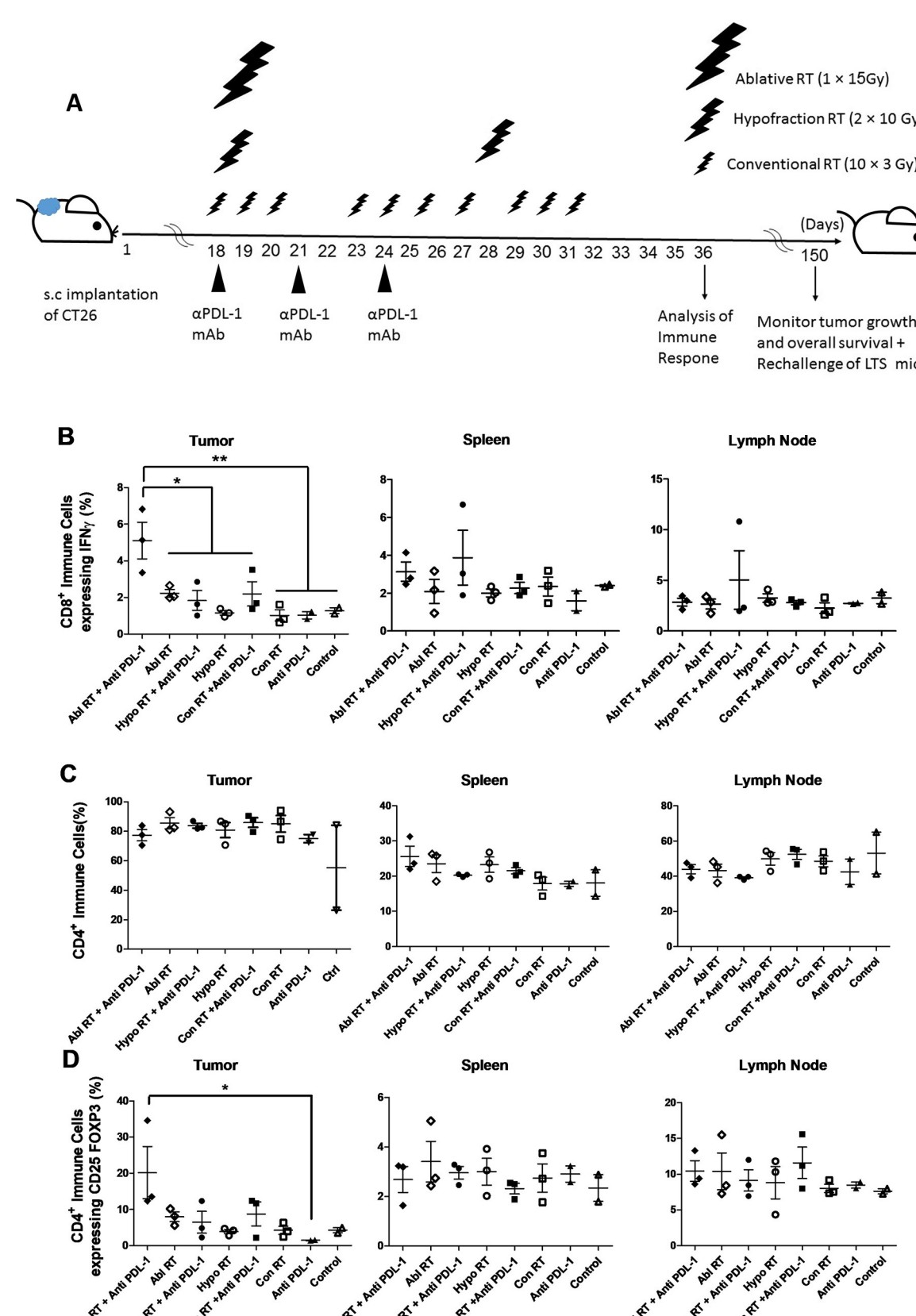

**Fig 2. Combining ablative radiation with αPD-L1 mAb increases CD8$^+$ effector T- and Treg cell infiltration in tumors.** The experimental schedule of Combining therapy of Ablative (n = 10), Hypofraction (n = 9) and Conventional (n = 10) Radiation regimens with αPD-L1 mAb are shown **(A)**. In combined therapy groups, mice received 200 μg/ml of αPDL-1 simultaneously with irradiation on day 18 post-initial inoculation and again on days 21 and 24. The percentage of CD8$^+$ IFNγ$^+$, CD4$^+$, and CD4$^+$ CD25$^+$ FOXP3$^+$ cells that infiltrated to tumor, spleen, and lymph node are shown respectively in **(B)**, **(C)** and **(D)**. Data are presented as means ± SDs and analyzed by Tukey's Multiple Comparison Test; (ns: non-significant*: P < 0.05, **: P < 0.01).

infiltration of CD8$^+$ cells expressing IFNγ (CD8$^+$ effector T-cell) and CD4$^+$ CD25$^+$ FOXP3$^+$ (Treg) cells to the tumor while hypofraction and conventional RT did not. The mean tumor volumes and percent survival of mice treated with the 3 regimens were not significantly different, likely due to them all receiving the same BED radiations (Fig 1C and 1D). These data demonstrate that infiltration of immune cells were differed when tumors were irradiated by different regimens with same BED given in different fractions and doses.

## Ablative RT combined with αPD-L1 mAb caused CD8$^+$ T cells and Treg cells to infiltrate into tumors in greater numbers than the other regimens

Ablative RT combined with αPD-L1 mAb led to a significant increase in the number of CD8$^+$ T cells expressing of IFNγ and Foxp3$^+$ CD25$^+$ expressing CD4$^+$ T cells infiltrating into the tumor, but not into spleen or lymph nodes (Fig 2B and 2D). The number of CD4$^+$ T cells did not change significantly in the other combined therapy groups (Fig 2C). These data demonstrate that ablative RT, when delivered in combination with αPD-1, leads to changes in tumor infiltration by CD8$^+$ effector T-cell and Treg populations.

## Ablative RT leads to IFNγ expression, and when combined with αPDL-1 mAb, significantly increased IFNγ expression in tumors, even in the long term after radiation

To determine whether infiltrated immune cells caused an adaptive change in tumors, the effector cytokine IFNγ was analyzed (Fig 3). We found that ablative radiation increased IFNγ expression in tumors in the long term after radiation relative to the control, while the other regimens decreased it insignificantly (Fig 3A). Also, ablative RT combined with αPDL-1 resulted in a 3-fold increase in IFNγ expression, while the other combination therapies had no different relative to their radiation monotherapies (Fig 3B). Histograms of IFNγ expression showing a shift to the right on the x-axis represent an increase in IFNγ expression on immune cells in the tumors (Fig 3C).

## Ablative RT is the most effective regimen for combining with αPDL-1 mAb at reducing tumor volume and increasing proportion of mice with complete tumor resolution and survival rates

Thirty days after therapy tumors were significantly smaller in all the irradiated mice than in controls, and this effect was even more pronounced when the irradiation was combined with αPDL-1. (Fig 4A and Table 1). The ratios of mice that experienced complete tumor resolution (Fig 4B) and survival rates (Fig 4C) were the same in these groups (Fig 4B). Tumor volumes were lowest and TTE and survival rates and the proportion of mice that experienced complete tumor resolution (6/7 mice) were greatest in mice that received the combining ablative RT with αPDL-1 (Fig 4B and 4C).

Although their BEDs were equal, the different irradiation regimens had different synergistic effects when combined with αPDL-1. In addition, tumor growth delay from ablative RT plus αPDL-1 was 89.23% greater than with ablative RT alone, hypofraction plus αAPD-L1

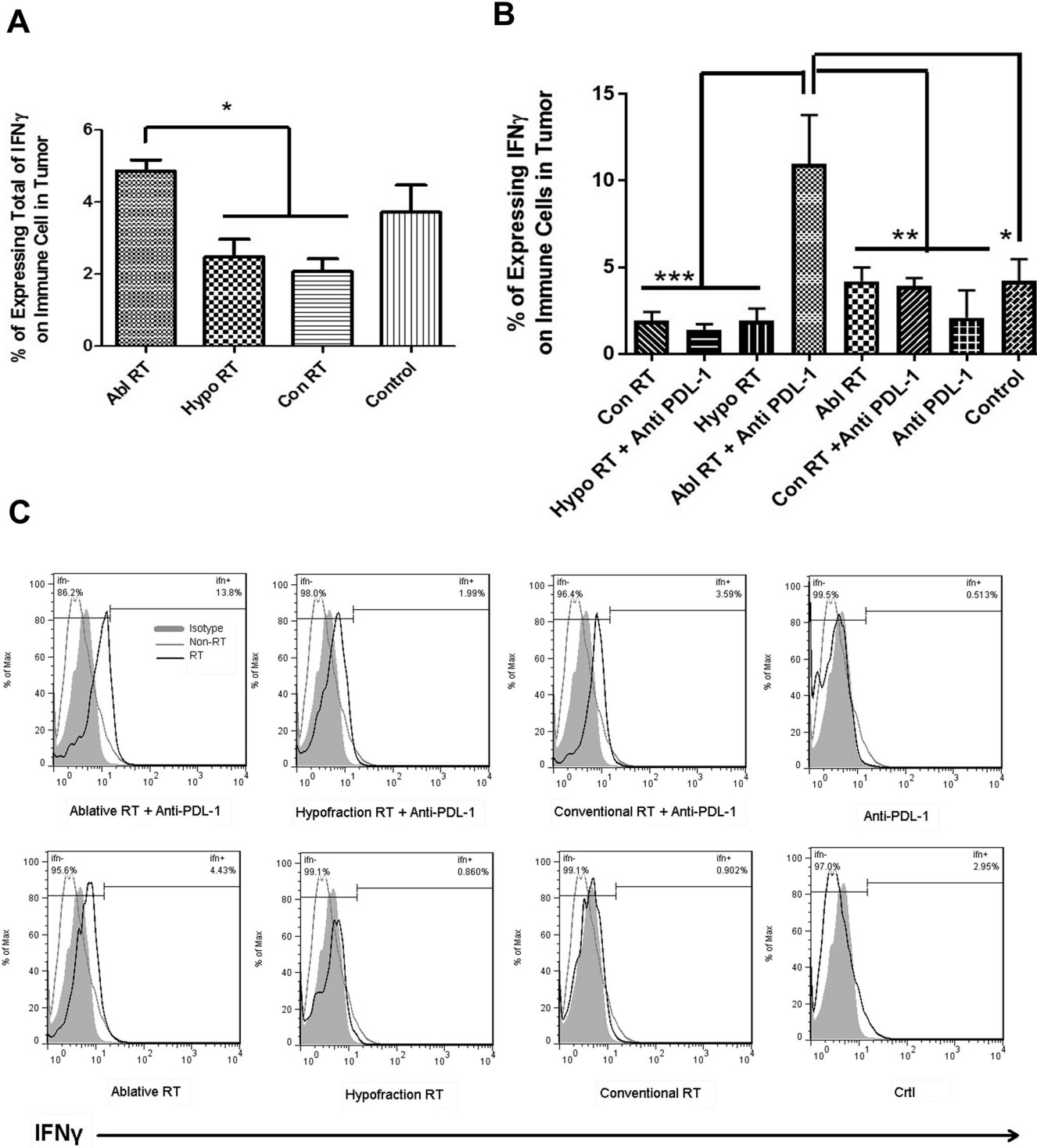

**Fig 3. Tumors from mice treated with ablative RT express significantly more IFNγ than tumors from mice treated with hypofractionated or conventional RT, and when ablative treatment combine with αPDL-1 mAb, tumors express significantly more IFNγ than those from mice treated with other regimens.** The percentage of expressing of IFNγ+ on immune cells in tumor are shown (**A**) in Ablative, Hypofraction and Conventional radiation and are shown in combining regimens of radiations with αPD-L1 (n = 3) (**B**). Histograms of IFNγ expression showing a shift to the right on the x-axis represent an increase in IFNγ expression on immune cells in the tumors (**C**). This chart was obtained using FlowJo 7.6.1 software. Data were analyzed by Tukey's Multiple Comparison Test and are presented as means ± SDs; ns: non-significant (*: P < 0.05, **: P < 0.01, ***: P < 0.001).

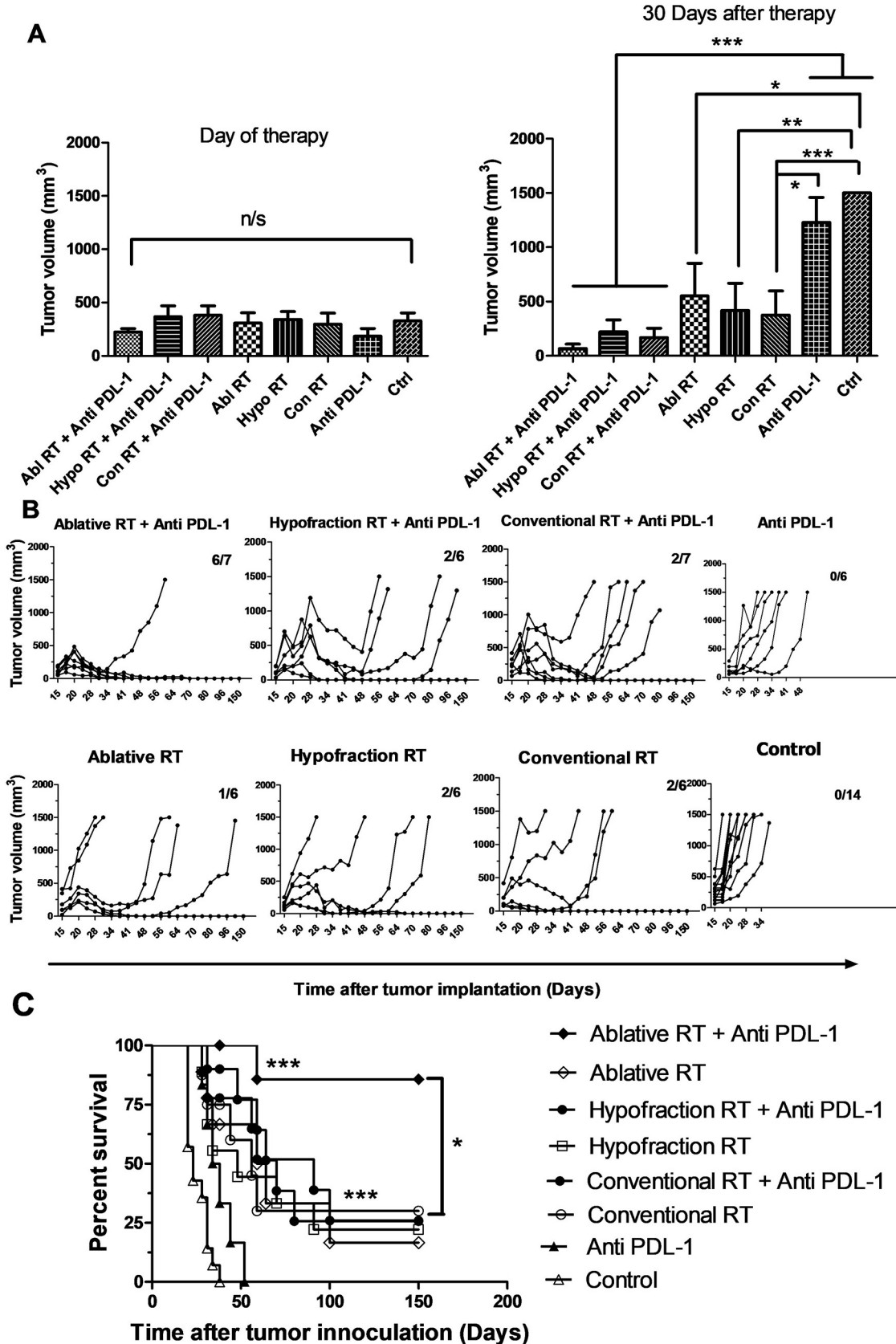

**Fig 4. When combined with αPDL-1 mAb, ablative radiation is the most effective regimen for reducing tumor volume, inducing complete tumor resolution, and increasing survival rates.** The average tumor volume in groups is shown at the onset of treatment and 30 days' post-treatment (at least n = 6) (**A**). On the day the treatment protocol began, tumor volumes were 300–400 mm$^3$ in all groups with no significant differences between groups. The tumor volume changes from the start of treatment to 150 days' post-treatment in each group are shown as Kaplan Meier curves (**B**). Survival rates are shown in (**C**). Combining Abl RT with αPDL-1 group had significant (P < 0.001) when compared with Control and αPDL-1 mice and had significant (P < 0.01) when compared with other therapies groups. ns: non-significant (*: P < 0.05, **: P < 0.01, ***: P < 0.001; Tukey's Multiple Comparison Test and Log-rank (Mantel-Cox) Test.).

was15.75% greater than hypofraction alone, and conventional RT plus αAPD-L1 7.57% greater than conventional RT alone.

## Treatment with ablative irradiation plus αPDL-1 mAb inhibit regrowth of new tumor, maybe due to creating a protective anti-tumor immune memory

We next investigated whether immunologic memory was generated in tumor-resolved-long term surviving (LTS) mice after treatment with each combination therapy at in vivo study. All the LTS mice that were treated with ablative (6/6) or hypofraction (2/2) RT plus αPDL-1 completely rejected tumors following re-challenge on days 90 and 150 post-initial challenge while 50% (1/2) of the LTS mice treated with conventional RT rejected tumors 150 days' post-initial challenge. The percent survival of LTS mice are shown in Fig 5 for 60 days after tumor challenge again. The LTS mice of Conventional RT had less ratio of survival time in compare of others therapy groups.

## Discussion

Our findings showed a significantly greater number of CD8$^+$ cells expressing IFNγ$^+$ in the TME after ablative radiation than with hypofraction or conventional RT. This finding was previously observed in high-dose irradiation studies [22, 23], and it was found that the expression of some immunogenic cell death (ICD) markers was increased when the radiation dose was greater than 5–10 Gy [24]. Although the BED was 40 Gy for all three regimens in our study, the different regimens had different effects on immune cell infiltration to the TME. This could lead to different responses between the regimens, either with or without αPDL-1.

**Table 1. The means of time to endpoint (TTE) the studied groups and the comparison of tumor growth delay (%TGD) between the studied and control groups, the studied and the αPD-L1 groups, and each combination therapy group with its corresponding RT group were calculated.**

| Groups | Average of TTE (Day) | % TGD Compared to Control Group | % TGD Compared to Anti-PDL-1 Group | % TGD of each Combination Therapy Group Compared to Radiotherapy Group |
|---|---|---|---|---|
| **Abl RT + Anti-PDL-1** | 137.12 | 433.70 | 279.96 | 89.23 |
| **Abl RT** | 72.46 | 182.035 | 100.79 | |
| **Hypo RT + Anti-PDL-1** | 99.79 | 288.39 | 176.51 | 15.75 |
| **Hypo RT** | 86.21 | 235.54 | 138.8 | |
| **Con RT + Anti-PDL-1** | 88.17 | 243.16 | 144.31 | 7.57 |
| **Con RT** | 81.96 | 219.00 | 127.11 | |
| **Anti-PDL-1** | 36.09 | 40.45 | 0 | |
| **Ctrl** | 25.69 | 0 | -28.80 | |

Abl RT: Ablative, RT: Radiotherapy, Hypo: Hypofraction, Con: Conventional, Ctrl: Control

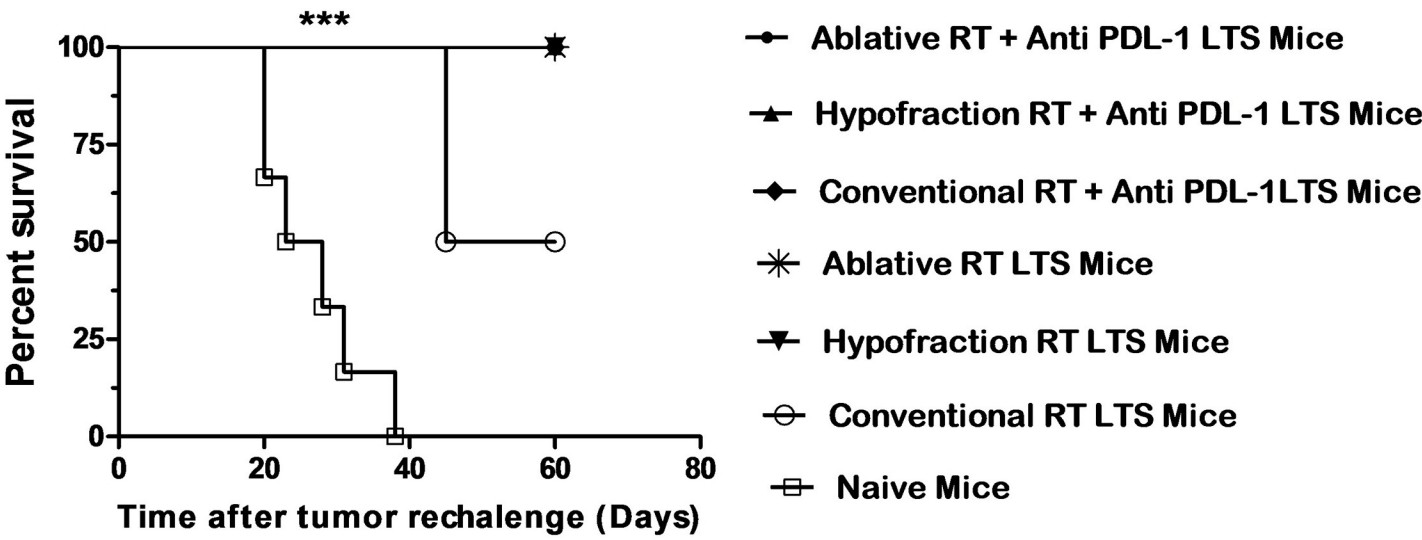

**Fig 5. Treatment with ablative irradiation plus αPDL-1 mAb inhibit regrowth of new tumor, maybe due to creating a protective anti-tumor immune memory.** The curve shows the survival rate in LTS mice (n = 2) for 60 days after tumor re-challenge again with $1 \times 10^6$ CT26 cells into the left flanks (another side) of mice. The survival rate in mice treated with ablative RT plus αPDL-1 was significantly greater than that of naïve mice (n = 5). The LTS mice of Conventional RT had less ratio of survival time in compare of others therapy groups (***: P <0.001).

The mean IFNγ expression in the conventional and hypofraction groups was less than in the control group; however, this difference was not significant. Murthy and colleagues stated that radiation-induced hypoxia can reduce IFNγ expression; as a result, RT is ineffective on the immune system in the tumor area [25].

It is therefore suggested that in further studies, the rate of hypoxia in the tumor area should be considered after irradiation with different regimens. Our study however, showed that IFNγ expression increased in the ablative group following the increase in infiltration of CD8+ cells into the tumor region after high-dose exposure. Another study found that when IFNγ expression by CD8+ cells increased in the tumor site after irradiation, this also increased PDL-1 expression on tumor cells, ultimately leading to exhausted CD8+ cells [26, 27], and so αPDL-1 could be considered an appropriate treatment [10].

We showed that αPDL-1 had synergistic effects when combined with single high-dose RT regimens, and of the regimens in this study, ablative irradiation stimulated IFNγ expression. It is likely that the expression of PDL-1 is different depending on the regimen in the radiated tumor region, just as IFNγ expression differed between regimens [28].

Also, this synergistic effect of high-dose RT with αPDL-1 was entirely TCD8+ dependent, as was shown in other studies [28, 29].

Our study also showed that the population of CD4+ FOXP3+ cells increased significantly in the high-dose single-radiation group with αPDL-1. Given that, unlike other immune cells, Treg cells were resistant to irradiation, Ratikan et al. compared several irradiation regimens with different BEDs and found that increasing the dose per fraction decreased the tumor volume, concurrent with an accumulation of Treg cells [23].

Another study showed that anti-PDL-1 enhanced the cytotoxic effects of IFNγ-dependent CD8+ cells [30]. However, another study in a CT-26 model found no significantly different effects between ablative (1 × 7Gy) RT, hypofraction (3 × 4Gy) RT, or conventional (5 × 2Gy) RT combination therapies with αPD-L1 [31]. We believe the reason for this discrepancy with our study is the difference BEDs in two studies. The total BEDs of the three regimens in their study were 11.2, 16.8 and 12 Gy, respectively. Subsequent meta-analysis studies indicate

increasing the irradiation BED could increase the likelihood of observing an immunological-dependent abscopal effect, and when the BED is 60 Gy, the probability of occurrence of this effect is 50% [17]. Low BED regimens are unlikely to stimulate immunity sufficiently; therefore, combining them with αPDL-1 has no additive effect. In another study, αPDL-1 was reported to have greater synergic effects when combined with ablative RTs ($8 \times 2$ Gy) than with conventional regimens ($10 \times 2$ Gy) [32].

Because the CT26 mouse model, in which the immune system is suppressed in TME, is more responsive to immune checkpoint blockade rather other models, it is appropriate for combined therapy with ICBs in pre-clinical studies, in addition to the T [33]. Recent studies suggested this tumor model is used to evaluate the synergistic effects of high-dose RT and ICBs [34], and other studies have suggested the use of the CT26 model due to its severely suppressed environment for immunological responses. The average tumor volume at the beginning of our study was 300–400 mm$^3$. This was largest from other studies [27], and could indicate the combination of ablative RT plus αPDL-1 is more effective than the other combination therapies in large tumor.

Different irradiation regimens with the same BED have equal effects on cell death, as the LQ model predicts [35]. In our study the three radiation regimens with the same BED resulted also in no differences in tumor size or survival rates, but attention should be paid to the immunological changes after irradiation from each of these regimens. For designing of combining immunotherapy with irradiation regimens, clinical researchers should consider how RT affects immunity, leading to effective planning for dose adjustment and number of fractions. Such clinical studies can ultimately accelerate the clinical development of RT regimens and their combination with immunotherapy, which ultimately leads to a strong and sustained immune response that eliminates the tumor and prevents recurrence.

## Supporting information

**S1 Checklist. The ARRIVE guidelines checklist.**
(PDF)

**S1 Fig.**
(TIF)

## Acknowledgments

This article has been extracted from the thesis written by Mrs. Maedeh Alinezhad in School of Medicine, Shahid Beheshti University of Medical Sciences. (Registration No:344)

## Author Contributions

**Conceptualization:** Mohsen Bakhshandeh, Nariman Mosaffa, Seyed Amir Jalali.

**Formal analysis:** Reza Alimohamadi.

**Funding acquisition:** Seyed Amir Jalali.

**Investigation:** Maedeh Alinezhad, Elham Rostami, Reza Alimohamadi, Seyed Amir Jalali.

**Methodology:** Mohsen Bakhshandeh, Seyed Amir Jalali.

**Project administration:** Maedeh Alinezhad, Seyed Amir Jalali.

**Software:** Seyed Amir Jalali.

**Supervision:** Nariman Mosaffa.

**Validation:** Seyed Amir Jalali.

**Writing – original draft:** Maedeh Alinezhad.

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
