## [Decision Letter · Decision Letter 0]

21 Jan 2020

PONE-D-19-33249

Synergistic Effects of Anti-PDL-1 with Ablative Radiation Comparing to other Regimens with Same Biological Effect Dose based on Different Immunogenic Response

PLOS ONE

Dear Dr Jalali,

Thank you for submitting your manuscript to PLOS ONE. After careful consideration, we feel that it is not fully meet PLOS ONE’s publication criteria. Therefore, we invite you to submit a revised version of the manuscript that addresses the points raised during the review process.

In addition, during our internal assessments, we noted that 6 animals died due to lowering temperature during the irradiation procedures.Some editors raised the great concerns about the animal welfare. We expect your response about this matter. 

We would appreciate receiving your revised manuscript by Mar 06 2020 11:59PM. To enhance the reproducibility of your results, we recommend that if applicable you deposit your laboratory protocols in protocols.io, where a protocol can be assigned its own identifier (DOI) such that it can be cited independently in the future. For instructions see: http://journals.plos.org/plosone/s/submission-guidelines#loc-laboratory-protocols

We look forward to receiving your revised manuscript.

Kind regards,

Jianxin Xue

Academic Editor

PLOS ONE

2. As part of your revision, please complete and submit a copy of the ARRIVE Guidelines checklist, a document that aims to improve experimental reporting and reproducibility of animal studies for purposes of post-publication data analysis and reproducibility: https://www.nc3rs.org.uk/arrive-guidelines. Please include your completed checklist as a Supporting Information file. Note that if your paper is accepted for publication, this checklist will be published as part of your article.

4. We note that Figures in your submission contain copyrighted images. All PLOS content is published under the Creative Commons Attribution License (CC BY 4.0), which means that the manuscript, images, and Supporting Information files will be freely available online, and any third party is permitted to access, download, copy, distribute, and use these materials in any way, even commercially, with proper attribution. For more information, see our copyright guidelines: http://journals.plos.org/plosone/s/licenses-and-copyright.

a)    You may seek permission from the original copyright holder of the Figures to publish the content specifically under the CC BY 4.0 license.

Reviewers' comments:

Reviewer's Responses to Questions

**Comments to the Author**

1. Is the manuscript technically sound, and do the data support the conclusions?

Reviewer #1: Yes

Reviewer #2: Yes

2. Has the statistical analysis been performed appropriately and rigorously? 

Reviewer #1: Yes

Reviewer #2: Yes

3. Have the authors made all data underlying the findings in their manuscript fully available?

Reviewer #1: Yes

Reviewer #2: Yes

4. Is the manuscript presented in an intelligible fashion and written in standard English?

Reviewer #1: Yes

Reviewer #2: Yes

5. Review Comments to the Author

Reviewer #1: Reviewer’s comments

The current study has evaluated the synergistic effect of RT combined with

immunotherapy using the αPDL-1 antibody in three different RT regimens in a xenograft tumor model in BALB/c mice. The synergistic effect of the αPDL-1 antibody in RT regimens is interesting. The findings would be helpful to develop the new immunomodulator and to design the more efficient radiation regimens to control the tumor outgrowth.

Comments to the author

1. In the Introduction section, the author should favorably explain the motivation of the research to differentiate the current study from the other applied papers in this field.

2. The author should explain more about the expression of CD8+ and CD4+ cells after radiation in the first paragraph of the introduction.

Methods:

3. The author should mention the fractionation schedules. How many fractions of RT were given per day and for how many days/ weeks?

4. The mostly used hyper fractionation RT regimens in clinics are 2Gy fraction per day. Is there any specific reason to use 3Gy fractions?

5. Radiation with or without causes nephrotoxicity. The author should evaluate the nephrotoxicity after RT ± anti-PD-L1 and could cite the following paper (PMID: 27836988)

Results and discussion

6. The quality of the figures is unsatisfactory, please improve the quality of the figures.

7. The findings have been discussed to some extent to support the claims made in the hypothesis. The manuscript adds some critical information to advance the field. The results presented in the paper are substantial enough for publication in Plosone after minor revision.

Reviewer #2: Excellent paper. Please add a paragraph after the discussion: Conclusion and discuss clinical implications of your findings and how your findings could be translated into the clinical studies. Lung SBRT is a good example where we deliver ablative dose of radiation.

6. PLOS authors have the option to publish the peer review history of their article (what does this mean?). If published, this will include your full peer review and any attached files.

Reviewer #1: Yes: ANIS AHMAD

Reviewer #2: No

---

## [Author Response · Author response to Decision Letter 0]

9 Mar 2020

February 1, 2020

Prof. Jianxin Xue

Academic Editor

PLOS ONE

Dear Professor Jianxin Xue:

Thank you very much for your letter regarding our manuscript (PONE-D-19-33249) entitled" Synergistic Effects of Anti-PDL-1 with Ablative Radiation Comparing to other Regimens with Same Biological Effect Dose based on Different Immunogenic Response”. The reviewers’ comments were very helpful, and we made revisions to the manuscript according to their criticisms.

I think described point together with the other minor points that we have addressed made the manuscript much improved. Please find attached a detailed reply to the referees’ comments and description of other changes to the manuscript.

We would like to thank the reviewers again for their important comments. Our responses to the comments are as follows:

#Academic editor: In addition, during our internal assessments, we noted that 6 animals died due to lowering temperature during the irradiation procedures. Some editors raised the great concerns about the animal welfare. We expect your response about this matter. 

Response: We appreciate you taking the time and the valuable suggestions offered. The mice were adapted to the conditions after two weeks in the animal research room. Then, after tumor induction, eight groups were randomly selected.

At the stage of tumor induction and irradiation, they were anesthetized to reduce pain tolerance. Symptoms of pain were monitored for up to 4 hours after irradiation, and if any symptoms of pain such as paleness, anorexia, weight loss and back flexion were observed, they were given pain relief. After reaching the final stage and finding endpoint criteria, they were anesthetized by isoflurane and sacrificed ten minutes later.

#Reviwer 1: 1. In the Introduction section, the author should favorably explain the motivation of the research to differentiate the current study from the other applied papers in this field.

Response: We thank the reviewer for their close attention. The motivations for the study were written in the form of a few questions at the end of the last paragraph, which we completed by adding a few sentences.

However, studies have not yet determined whether different regimens with the same BED (when the biological effects of the regimens are equal) have different immunogenic effects, whether they affect tumor recurrence associated with immunological responses. In addition, can these regimens respond differently to combination therapy? Because it can improve the quality of immune responses into TME by use of modification in the dose and number of fractions radiation to the extent that the radiation clinic limitations and technology of the radiation devices allow us. Also it can lead to memory responses, which prevents tumor recurrence that is a common problem after treatment. To address these issues, we aimed to evaluate the synergistic effect of RT combined with immunotherapy using αPDL-1 in three different RT regimens with the help of modeled tumor BALB/c mice.

#Reviwer 1: 2. The author should explain more about the expression of CD8+ and CD4+ cells after radiation in the first paragraph of the introduction. 

Response: Yes, Sure, in the first paragraph, we noted the effect of radiation on CD4 and CD8 cells.

Irradiation has multiple inhibitory and stimulatory effects on the immune system such as increased CD4+, CD8+ and Treg Cells infiltration into TME (3, 4) Some regimens of radiation like ablative RT was associated with increased infiltration of exhausted CD8+ T cells into the tumor, which induced radiation resistance. (5, 6). The combination of radiotherapy and Immune checkpoint blockers therapy can reduce these inhibitory effects by stimulating immune cells and enhancing the response to radiation. Preclinical studies showed that RT increased PDL-1 expression on tumor cells (7, 8), and anti-PDL1 (αPDL-1) mAb combined with radiation had a synergistic effect on immune response induction dependent IFN-γ producing CD8 T cells activations. (9, 10).

#Reviwer 1: 3. The author should mention the fractionation schedules. How many fractions of RT were given per day and for how many days/ weeks?

Response: Thank you for pointing this out. In Methods, Treatment section adds these sentences:

The experimental schedule of ablative radiation therapy was only one fraction of 15 Gy in 18 days’ post-initial inoculation. In Hypofraction RT regimens, two fractions of 10 Gy were performed on days 18 and 28 after inoculation. In Conventional RT, ten fractions of 3 Gy were given from day 18 to day 31 for 14 days at a time interval of one day or at most two days. (Figure 1A).

#Reviwer 1: 4. The mostly used hyper fractionation RT regimens in clinics are 2Gy fraction per day. Is there any specific reason to use 3Gy fractions?

Response: We thank the reviewer for their close attention. Given that the clinic conventional RT has a range of 1.8 to 2.5 Gy1, we wanted to assay immunological responses when deliver a 40 Gy equivalent total BED in these regimens. But on the other hand, we were going to have to choose the 2g dose for conventional regimen, we would have to give 17 fractions of radiation, due to the limitation of the number of irradiations in the pre-clinical studies, 3Gy fractions were selected for 10 fractions. Of course, further studies are ongoing for lower dose of BED regimens closer to the clinic regimens.

#Reviwer 1: 5. Radiation with or without causes nephrotoxicity. The author should evaluate the nephrotoxicity after RT ± anti-PD-L1 and could cite the following paper (PMID: 27836988)

Response: The main purpose of this study was to evaluate the immunological responses in irradiated regimens with the same biological effects. Further studies on radiation induced nephropathy should be considered after identifying the most appropriate regimen in any body tissue according to the immunological response and combination of radiation therapy with immunotherapy.

#Reviwer 1: 6. The quality of the figures is unsatisfactory, please improve the quality of the figures. 

Response: The correction was made in the revised manuscript. The quality of figures are improved.

#Reviwer 1:7. The findings have been discussed to some extent to support the claims made in the hypothesis. The manuscript adds some critical information to advance the field. The results presented in the paper are substantial enough for publication in Plosone after minor revision.

Response: We greatly appreciate the reviewer’s efforts to carefully review the paper.

#Reviwer 2: Excellent paper. Please add a paragraph after the discussion: Conclusion and discuss clinical implications of your findings and how your findings could be translated into the clinical studies. Lung SBRT is a good example where we deliver ablative dose of radiation.

Response: We greatly appreciate the reviewer’s efforts to carefully review the paper.

References:

1. John. L. Meyer. IMRT, IGRT, SBRT: Advances in the Treatment Planning and Delivery of radiotherapy. Vol. 43. Page 396

---

## [Decision Letter · Decision Letter 1]

25 Mar 2020

Synergistic Effects of Anti-PDL-1 with Ablative Radiation Comparing to other Regimens with Same Biological Effect Dose based on Different Immunogenic Response

PONE-D-19-33249R1

Dear Dr. Jalali,

We are pleased to inform you that your manuscript has been judged scientifically suitable for publication and will be formally accepted for publication once it complies with all outstanding technical requirements.

With kind regards,

Jianxin Xue

Academic Editor

PLOS ONE

Additional Editor Comments (optional):

Reviewers' comments:

Reviewer's Responses to Questions

**Comments to the Author**

1. If the authors have adequately addressed your comments raised in a previous round of review and you feel that this manuscript is now acceptable for publication, you may indicate that here to bypass the “Comments to the Author” section, enter your conflict of interest statement in the “Confidential to Editor” section, and submit your "Accept" recommendation.

Reviewer #1: All comments have been addressed

2. Is the manuscript technically sound, and do the data support the conclusions?

Reviewer #1: Yes

3. Has the statistical analysis been performed appropriately and rigorously? 

Reviewer #1: Yes

4. Have the authors made all data underlying the findings in their manuscript fully available?

Reviewer #1: Yes

5. Is the manuscript presented in an intelligible fashion and written in standard English?

Reviewer #1: Yes

6. Review Comments to the Author

Reviewer #1: (No Response)

7. PLOS authors have the option to publish the peer review history of their article (what does this mean?). If published, this will include your full peer review and any attached files.

Reviewer #1: Yes: ANIS AHMAD

---

## [Editor Report · Acceptance letter]

30 Mar 2020

PONE-D-19-33249R1 

Synergistic Effects of Anti-PDL-1 with Ablative Radiation Comparing to other Regimens with Same Biological Effect Dose based on Different Immunogenic Response 

Dear Dr. Jalali:

I am pleased to inform you that your manuscript has been deemed suitable for publication in PLOS ONE. Congratulations! Your manuscript is now with our production department. 

With kind regards,

on behalf of

Dr. Jianxin Xue 

Academic Editor

PLOS ONE